# Green synthesized silver nanoparticles from *Moringa*: Potential for preventative treatment of SARS-CoV-2 contaminated water

Omorilewa B. Ebunoluwa[1], Adebayo J. Bello[2¤], Rukayat O. Ayorinde[2], Nneka Onyejepu[3], Joseph O. Shaibu[4], Adeniyi R. Adewole[5], Abeebat O. Adewole[6], Olusegun A. Adedeji[2], Ololade O. Akinnusi[7], Olajumoke B. Oladapo[1], Temitope S. Popoola[1], Oluwamodupe M. Arotiba[5], Joseph B. Minari[1], Luqman A. Adams[8], Joy Okpuzor[1], Mujeeb O. Shittu[9*]

1 Department of Cell Biology and Genetics, University of Lagos, Yaba, Lagos, Nigeria, 2 Department of Biological Sciences, Redeemer's University, Ede, Osun, Nigeria, 3 Centre for Tuberculosis Research, Nigerian Institute of Medical Research, Yaba, Lagos, Nigeria, 4 Centre for Human Virology and Genomics, Nigerian Institute of Medical Research, Yaba, Lagos, Nigeria, 5 Chevron Molecular Biology Research Laboratory, College of Medicine, University of Lagos, Yaba, Lagos, Nigeria, 6 Department of Biomedical Science, Teesside University, Middlesbrough, United Kingdom, 7 Department of Nursing and Community Health, Glasgow Caledonian University, Glasgow, United Kingdom, 8 Department of Chemistry, University of Lagos, Yaba, Lagos, Nigeria, 9 Department of Biotechnical and Clinical Laboratory Science, Jacobs School of Medicine and Biomedical Sciences, The State University of New York (SUNY), University at Buffalo, Buffalo, New York, United States of America

¤ Current Address: School of Pharmacy and Biomolecular Sciences, Liverpool John Moores University, Liverpool, United Kingdom
* mujeebsh@buffalo.edu

## Abstract

Biogenic silver nanoparticles have been reported as good antimicrobial candidates. In this study, we synthesized and characterized silver nanoparticles using aqueous leaf extracts of *Moringa oleifera* (AgNPmo) and investigated their antimicrobial and antiviral activities. The AgNPmo demonstrated antibacterial activity against *Pseudomonas aeruginosa* and *Staphylococcus aureus*, with concentration-dependent zones of inhibition ranging from 6.5–15.5 mm and 0–7 mm, respectively. Cytotoxicity was assessed on Vero cells using a CCK-8 assay, which revealed an IC50 value of 38 µg/ml, indicating relatively low toxicity at effective concentrations. The antiviral activity against SARS-CoV-2 was confirmed using quantitative RT-PCR: AgNPmo exposure led to a time- and dose-dependent increase in Ct values for ORF1ab and N genes, with the strongest inhibition observed after 48 h. These results provide direct evidence of both antimicrobial and antiviral activity. The green synthesis using *Moringa* extracts offers added advantages by employing phenolic and flavonoid compounds as natural reducing and capping agents, making the process eco-friendly and biocompatible. While direct wastewater treatment was not tested, these findings highlight the advantages of Moringa leaf extract as a natural reducing and capping agent that enabled rapid, eco-friendly AgNP formation, while the resulting AgNPmo

**Data availability statement:** All relevant data are within the manuscript and its Supporting Information files.

**Funding:** The author(s) received no specific funding for this work.

**Competing interests:** The authors have declared that no competing interests exist.

demonstrated antimicrobial and antiviral activity. Thus, AgNPmo represents a promising, sustainable option for point-of-use disinfection and potential environmental water treatment.".

## Introduction

Severe acute respiratory syndrome coronavirus-2 (SARS-CoV-2) is the causative agent of a contagious, life-threatening disease called Coronavirus-2019 (COVID-19) [1,2]. So far, SARS-CoV-2 has spread globally to all continents since the World Health Organization (WHO) proclaimed it a Public Health Emergency of International Concern (PHEIC) in January 2020, with more than 775 million illnesses and more than 7 million deaths confirmed as of May 2024 according to COVID-19 dashboard on World Health Organisation website. The challenges and impacts of the disease on human health, economy, and environment have led to a plethora of studies that are focused on establishing ways of curtailing the various transmission routes of the COVID-19 virus [3].

One of the undermined transmission routes is by water contaminated with virus-infected human bodily excreta [4], and the virus is found to persist for seven days at 23°C [5]. SARS-CoV-2 virus prevalence in wastewater has been documented to pose a risk for the transmission of COVID-19 [6,7]. Also, the virus has been found in the human gastrointestinal system [8], which can be shed via faeces and may subsequently find its way into water bodies, especially from medical wastewater, wastewater from cruise ships and aircraft that are not properly managed [9]. This can lead to a developing problem, as water bodies or their components may serve as an overlooked medium for transmission of the SARS-CoV-2 virus and other viral infections. This study attempts to address this problem in a cost-effective way and could serve as a potential solution to eradicate SARS-CoV-2 in water bodies, therefore halting or slowing down the infection rate [10].

Silver nanoparticles (AgNPs) have become significantly important among other metal nanoparticles (NPs) due to their distinctive size, morphology, and environment-dependent properties, which are different from other materials' bulk properties [11,12]. AgNPs have been used extensively as antimicrobial agents. Different studies have reported the properties and efficacy of these AgNPs against bacteria [13–15], fungi [16,17], and viruses [18–20]. The efficacy of AgNPs against microbes is due to the physicochemical properties of AgNPs, such as small particle size and high surface-area ratio, which enable their movement through cellular membranes to the target sites [18,21,22]. Also, the easy movement of AgNPs into living cells causes accumulation, which can lead to toxic effects at very low concentrations [23]. Hence, the biosynthesis of the nanoparticles provides eco-friendly and bio-compatible alternatives for microbial treatments.

This study focuses on the use of AgNPs biosynthesized from *Moringa oleifera* as a disinfectant for treating SARS-COV-2-infected water. *Moringa oleifera* Lam (drumstick tree), belonging to the family *Moringaceae*, is among the most useful medicinal trees in most of Asia and Africa. *Moringa oleifera* leaf extracts possess biocompounds with

antioxidant and antibacterial activities [24]. These compounds act against the growth of gram-positive and gram-negative bacteria [25]. The biocompounds extracted from *Moringa oleifera* leaves also act as reducing and capping agents in the synthesis of silver nanoparticles [14], making them attractive antimicrobial agents. Moringa extracts have demonstrated efficacy in synthesizing metallic nanoparticles, providing a nontoxic and biocompatible approach [26]. Different parts of the Moringa plant, including leaves, seeds, and bark, possess a diverse array of bioactive compounds, making them suitable for green synthesis applications. The plant's inherent medicinal properties can enhance the antimicrobial and anticancer properties of nanoparticles, highlighting the potential of utilizing medicinal plants in the synthesis of nanomaterials [27]. The use of Moringa extracts in the synthesis of silver nanoparticles holds particular promise for water treatment applications due to the plant's inherent antimicrobial properties and wide availability. Green synthesis using plant extracts is a unique, systematic, affordable, and environmentally sound method for synthesizing nanoparticles with selective and specific properties and applications [28]. This is achieved through single-step processing, which is advantageous over chemical methods that use harmful and toxic chemicals [29,30]. Plant-based synthesis is environmentally benign because it uses less energy and creates less waste.

Moringa extracts have demonstrated efficacy in synthesizing metallic nanoparticles, providing a nontoxic and biocompatible approach [26]. Different parts of the Moringa plant, including leaves, seeds, and bark, possess a diverse array of bioactive compounds, making them suitable for green synthesis applications. The plant's inherent medicinal properties can enhance the antimicrobial and anticancer properties of nanoparticles, highlighting the potential of using medicinal plants in synthesizing nanomaterials [27]. The use of Moringa extracts in the synthesis of silver nanoparticles holds particular promise for water treatment applications due to the plant's inherent antimicrobial properties and wide availability. Green synthesis using plant extracts is a unique, systematic, affordable, and environmentally sound method for synthesizing nanoparticles with selective and specific properties and applications [28]. This is achieved through single-step processing, which is advantageous over chemical methods that use harmful and toxic chemicals [29,30].

Furthermore, plant-based synthesis is environmentally benign because it uses less energy and creates less waste. The green synthesis of silver nanoparticles using Moringa extracts offers a sustainable and environmentally friendly approach to water disinfection, with potential applications in point-of-use water treatment systems and large-scale water purification plants. Silver nanoparticles were successfully synthesized in this study using aqueous leaf extracts of the plant *Moringa oleifera,* and characterized using UV-Vis spectroscopy, Fourier transform infrared spectroscopy (FTIR), scanning electron microscopy (SEM), and X-ray diffractometry (XRD). The successful testing of the silver nanoparticles (AgNPs) was first carried out on clinical bacterial isolates to ascertain their antimicrobial properties. In this study, the AgNPmo showed a dose-and-time-dependent antiviral activity against SARS-CoV-2, showing that the AgNPmo has the potential to be used for the purification or treatment of SARS-CoV-2 contaminated water without exhibiting toxicity against living cells.

## Results and discussion

The search for effective treatments and preventive measures against SARS-CoV-2 is still ongoing, and continuous progress in this search is pertinent to minimize the spread of the virus in water and water bodies [5,31]. Silver nanoparticles (AgNPs) have recently been studied as potential antiviral agents due to their unique properties [32]. They have been shown to exhibit antiviral activities against several viruses, including influenza [33], HIV [34], and SARS-CoV-2 viruses [20].

Thus, this study aimed to assess the effectiveness of dose-dependent biogenic silver nanoparticles against SARS-CoV-2 in an aqueous environment over a period of time. The silver nanoparticles (AgNPmo) were synthesized using *Moringa oleifera* and characterized using UV-Vis spectroscopy, FT-IR, scanning electron microscopy (SEM), Energy-dispersive X-ray spectroscopy (EDX), and X-ray diffraction. The AgNPmo was first tested against clinical isolates of *Staphylococcus aureus* and *Pseudomonas aeruginosa* to ascertain its efficacy. Then, a cytotoxicity test was conducted on VeroE6 cells to determine the IC50 for the subsequent antiviral assay. The antiviral activity of AgNPmo against SARS-CoV-2 was conducted using qPCR assay in a dose and time-dependent manner.

## UV-visible analysis of the AgNPmo

The AgNPmo synthesis was first carried out using *Moringa oleifera* leaf extracts. Previous reports have shown that *M. oleifera* can serve as a reducing and capping agent in the synthesis of silver nanoparticles [35,36], and the derived nanoparticles can act effectively against bacteria and viruses [14,36,37]. In our study, the synthesis of AgNPmo was first ascertained visually by the color change of the reaction mixture, changing from yellowish to dark brown within 5 minutes (Figure in S2 Fig). The UV-Vis spectroscopy revealed a characteristic absorption peak at 420 nm (red line), indicating the formation of AgNPmo (Fig 1).

The observed absorbance peak wavelength is typical of silver nanoparticles, which falls within the range of 400–450 nm [38]. The spectrum (blue line) showing a peak at 410 nm was taken after 12 months of nanoparticle synthesis. This suggests that the AgNPmo retained its absorption properties over a long period of time. Therefore, in terms of nanoparticle stability and synthesis longevity, our UV-Vis spectral consistency over 12 months correlates with reports by Asif et al. (2022) [36], who highlighted the extended shelf-life and consistent absorbance properties of plant-derived AgNPs synthesized using Moringa oleifera. Similarly, previous studies have observed agglomeration in SEM images without loss of biological activity, a phenomenon mirrored in our findings and suggesting the functional resilience of biosynthesized nanoparticles despite morphological heterogeneity Moodley et al. (2018) [35]. The molecular interaction of the synthesized nanoparticles could be attributed to the surface plasmon resonance (SPR) due to the interaction of the free electrons found in metal-based synthesized nanoparticles with light energy [39].

The FTIR analysis in Fig 2 revealed the functional groups corresponding to absorbance peaks, which are interpreted according to the spectra correlation table [40]. The peaks, wavenumber (cm⁻¹), and their indication are represented in Table 1. The spectra suggest that the biomolecule compounds in the *M. Oleifera* leaf extract act by reducing the Ag ions through interaction with biomolecule functional groups and thus function as capping agents in the formation of the AgNPmo within a uniform size and shape [41]. Additionally our FTIR data confirm the presence of functional groups associated with phenolics and flavonoids from *Moringa*, supporting previous findings by Mohammed and Hawar (2022) [41], who demonstrated that plant metabolites play an essential role in both nucleation and stabilization of AgNPs. This contributes to uniformity and enhances the biological activity of the synthesized nanoparticles.

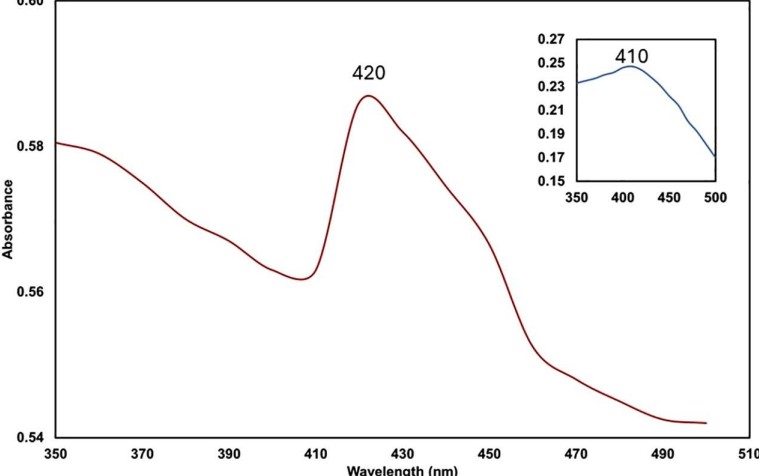

**Fig 1. Spectrophotometry analysis of the biosynthesized silver nanoparticles.** The analysis immediately after the synthesis of AgNPmo (red line) and analysis after 12 months of synthesis (blue line) show similar peaks, indicating the long-term stability of AgNPmo. A major absorption peak at 420 nm corresponds to the AgNP surface plasmon resonance.

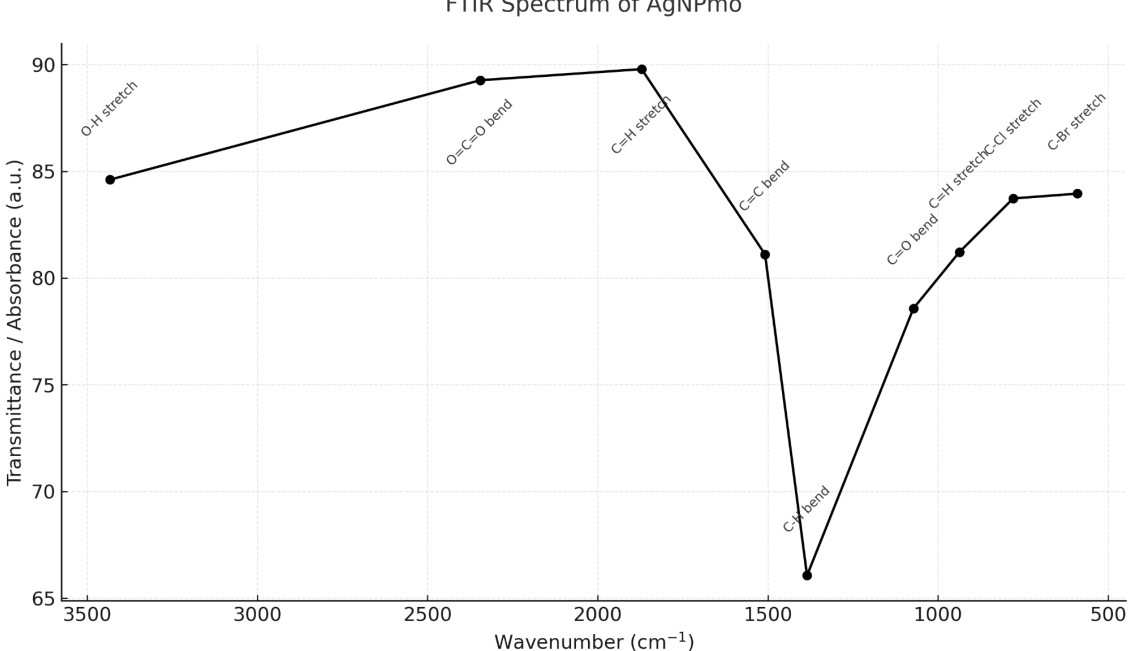

**Fig 2. FTIR spectrum of biosynthesized silver nanoparticles (AgNPmo). Major absorption bands were observed as stated in Table 1.** These functional groups suggest the presence of phytochemicals from Moringa oleifera extract that act as reducing and stabilizing agents during nanoparticle synthesis.

**Table 1. FTIR analysis of the biosynthesized silver nanoparticles.**

| Peak | Intensity | Bond/Stretching | Functional Group |
|------|-----------|-----------------|------------------|
| 590.24 | 83.953 | C-Br stretch | Alkyl halides |
| 779.27 | 83.728 | C-Cl stretch | Amide group |
| 937.44 | 81.213 | C=H stretch | Alkene |
| 1072.46 | 78.574 | C=O bend | Carbonyl group |
| 1384.94 | 66.062 | C-H bend | Alkane |
| 1508.38 | 81.109 | C=C bend | Aromatic compound |
| 1871.01 | 89.787 | C=H stretch | Aromatic compound |
| 2345.52 | 89.265 | O=C=O bend | Carbon dioxide |
| 3433.41 | 84.606 | O-H stretch | Hydroxyl group |

## SEM analysis of the AgNPmo

The SEM analysis provided an inconclusive evaluation of the shape and size of the AgNPmo synthesized. Agglomeration of the nanoparticles was observed and associated with the purification procedure during synthesis. These results are similar to the findings of Moodley et al. (2018) [35].

The SEM analysis identifies the morphology and particle size of the nanoparticles using the microscopy technique. In Fig 3, the AgNPmo presents an irregular surface topography with different shapes and sizes. The AgNPmo was observed at 25,000 g magnification on a 3μm scale. Most of the particles aggregated, which could result from the preparation process for SEM analysis. Some of the particles (in red circles) are less than 500nm, which may indicate that tiny nanoparticles are within the agglomerated particles. SEM analysis revealed agglomerated yet functional particles, an issue

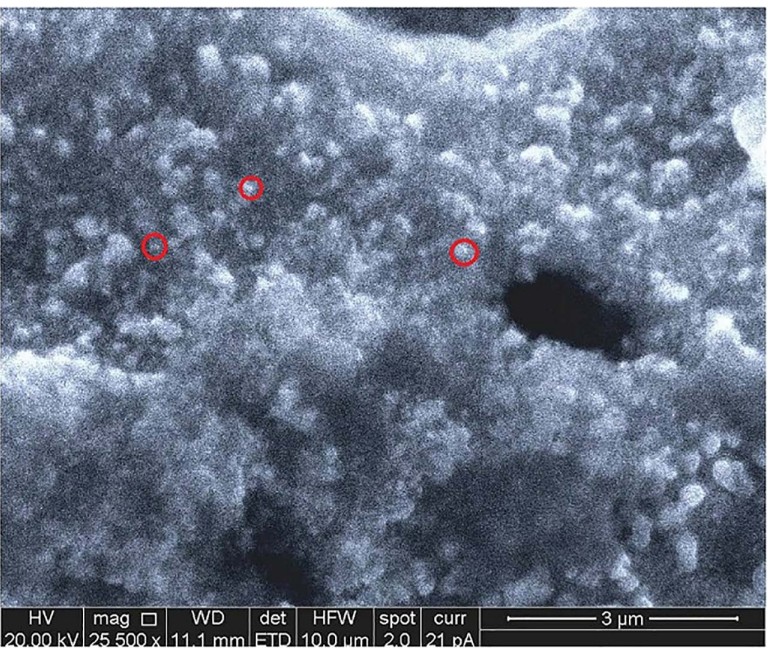

**Fig 3. Scanning electron microscopy showing the particle size of the biosynthesized nanoparticles.** Red circles indicate representative particles; however, due to agglomeration, size distribution was further supported by XRD analysis.

frequently encountered in green synthesis methods [35] and more recently Menichetti et al. (2023) [15], emphasized that despite morphological irregularities, agglomerated AgNPs can retain strong antimicrobial activity due to their high surface reactivity and embedded functional phytochemicals.

## XRD analysis of the AgNPmo

The XRD analysis is a good indication of the crystalline structure and stability of nanoparticles. The AgNPmo synthesized, confirmed by the XRD pattern (Fig 4), showed diffraction spectra of 27.91°, 32.37°, and 38.30°. 44.41°, 46.20°, thus revealing the crystalline nature of the synthesized nanoparticles [42]. The diffractogram peaks at 32.37°, 38.30°. 44.41° corresponds to peaks (100), (111), and (200) crystallographic planes when compared with the standard powder diffraction card of the Joint Committee on Powder Diffraction Standards (JCPDS) (Fig 4) [43].

The crystalline nature of our nanoparticles is critical because crystallinity has been correlated with enhanced biological reactivity. Similar crystalline diffraction patterns have been noted in comparable studies on biosynthesized AgNPs used for antimicrobial and antiviral applications [20].

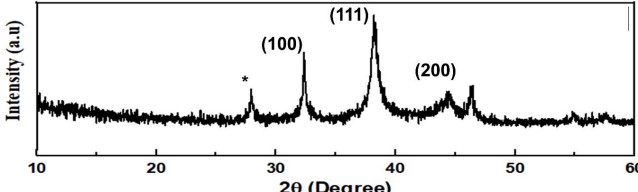

**Fig 4. XRD analysis of the biosynthesized nanoparticles.**

## Analysis of antimicrobial activity

Exposure of clinical bacterial pathogens to AgNPs has been reported to disrupt membrane permeability, prompting leakage from cells and hindering their growth and replication [44–46]. First, to confirm the efficacy of the synthesized silver nanoparticles, the antibacterial activities of the AgNPmo were carried out on *Pseudomonas aeruginosa* and *Staphylococcus aureus* clinical isolates (Fig 5A and 5B). The effectiveness of AgNPs extracted from *Moringa oleifera* against the tested bacterial strains was evident in the susceptibility *of P. aeruginosa*, showing a notable sensitivity with a mean zone of inhibition (ZOI) diameter of 15.5, 11, 7.5, and 6.5 mm at AgNPmo concentrations of 100% (1.68 mg/mL), 50, 25 and 12.5%, respectively (Fig 5C). Similarly, for *S. aureus,* a lesser mean ZOI diameter of 7.0, 4, 3.5, and 0 mm at similar concentrations of the AgNPmo, respectively (Fig 5D). The NC (negative control) containing DMSO without the AgNPmo showed no effect on the bacterial isolates, confirming the antimicrobial activity of the synthesized nanoparticles (Figure in S1 Fig). The result suggests a direct correlation between the quantity of AgNPs and the expansion of the ZOI. In agreement with our study, some reports have documented the action of AgNPs against bacterial proliferation in a concentration-dependent manner [47,48]. The AgNPmo demonstrated dose-dependent inhibition against *P. aeruginosa* and *S. aureus*, with the former showing greater susceptibility. This is consistent with reports by Ferreres et al. (2023) [48], who found

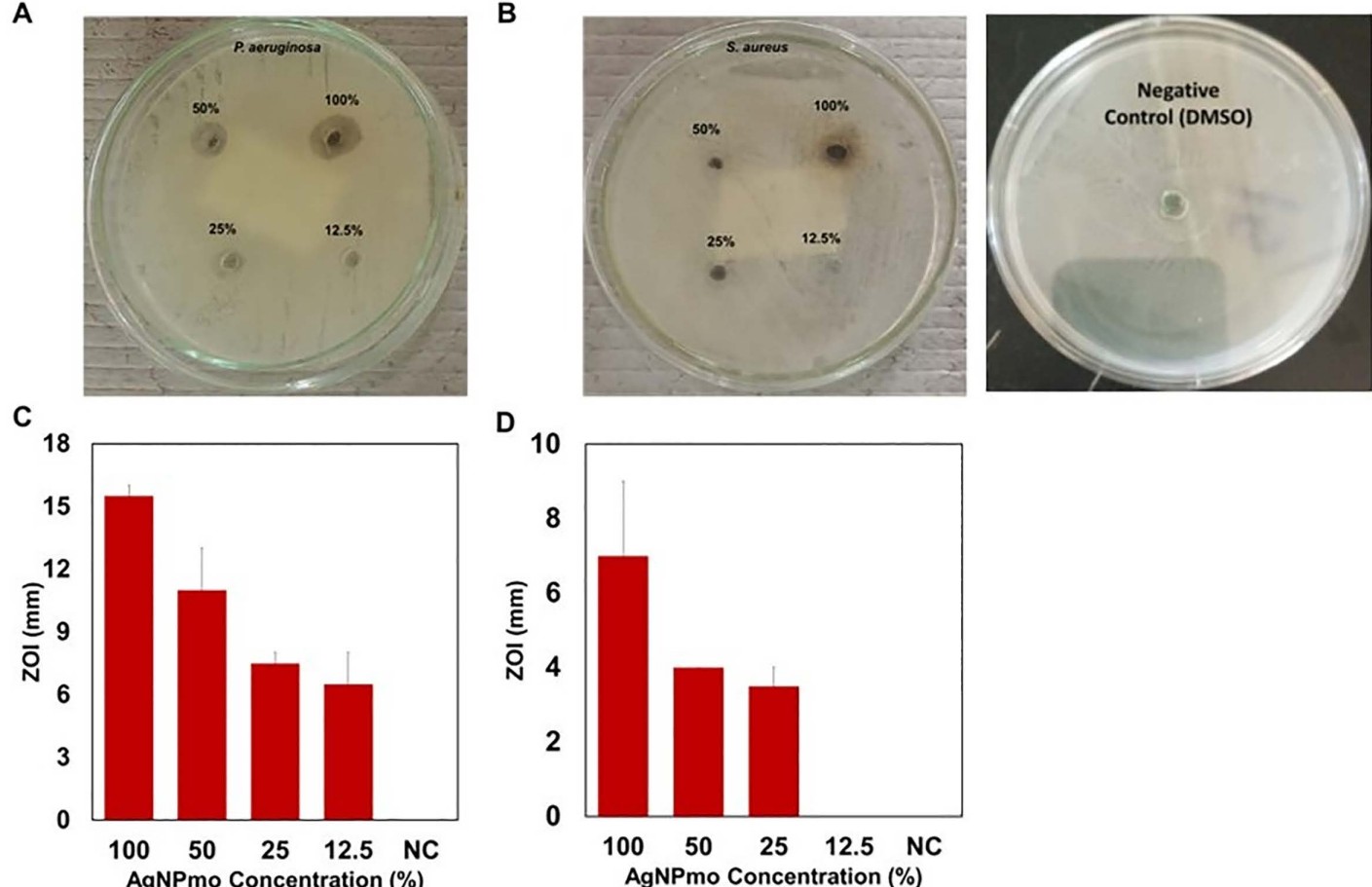

**Fig 5. Antibacterial activities of the AgNPmo.** Zone of Inhibition (ZOI) of *P. aeruginosa* and *S. aureus*, respectively, at different concentrations of the AgNPmo in (5A) and (5B). Dose-dependent increase in ZOI for *P. aeruginosa (5C)* and *S. aureus (5D)*. Negative control (NC) contains DMSO only.

that Gram-negative bacteria, due to thinner peptidoglycan layers, are more vulnerable to the oxidative stress induced by AgNPs. Ultimately, AgNPs interfere with bacterial macromolecules, causing breakdown and eventual cell death. Our observed trend—larger inhibition zones with increasing AgNPmo concentrations—parallels the concentration-dependent action described by Tripathi and Goshisht (2022) [44].

**Cytotoxicity assay**

After confirming the efficacy of the AgNPmo by evaluating its antimicrobial activity against clinical bacterial isolates, the cytotoxicity effect on VERO cells was evaluated to determine the inhibitory concentration ($IC_{50}$) for the subsequent antiviral study. The results revealed that the AgNPmo exhibited significant cytotoxic activities in a dose-dependent manner (Fig 6).

The statistical analysis performed on data derived from the CCK-8 assay revealed an IC50 value of 38 μg/mL for the 10mM stock concentration of the AgNPmo compared to the control, indicating relatively low cytotoxicity at effective concentrations. At higher doses (> 312.5 μg/mL), the assay showed negative viability values, reflecting high cytotoxicity. These findings suggest that while AgNPmo is biocompatible at low to moderate concentrations, excessive doses can disrupt cell integrityThe viability of untreated Vero cells (control) remained unchanged at 100% cell viability. The $IC_{50}$ value in this study is lower compared to a previous study, which reported a 568 g/mL $IC_{50}$ of AgNPs biosynthesized using *Catharanthus roseus* [49]. The low $IC_{50}$ value indicates a minimal inhibitory effect on the viability of Vero cells, making AgNPmo biocompatible with human cells. Vero cells were used for cytotoxicity assays because they are a well-established model for preliminary screening of nanoparticle toxicity and are routinely used in SARS-CoV-2 research. Direct cytotoxicity assays on SARS-CoV-2-infected cells require BSL-3 facilities, which were beyond the scope of this study. This finding supports the potential of synthesized AgNPmo for safe environmental applications, especially in water treatment scenarios.

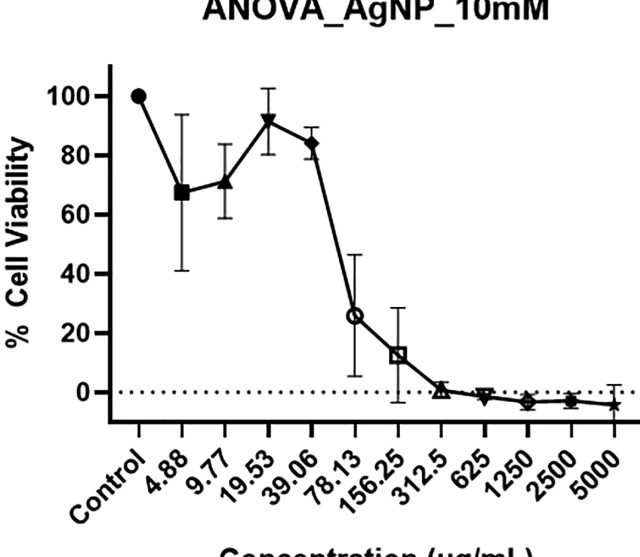

**Fig 6. Percentage cell viability Vero cells treated with the 10mM biosynthesized silver nanoparticles at different concentrations.** Cells were cultured for 24 h prior to treatment. At high concentrations (>312.5 μg/mL), AgNPmo caused marked cytotoxicity.

## Quantitative RT-PCR analysis

Quantitative PCR (qPCR) assay is a primary method of detecting the viral load of SARS-CoV-2 in various clinical specimens, including nasopharyngeal swabs stored in viral transport medium, sputum, and saliva [50]. The qPCR assay targets the viral RNA, which is extracted from the clinical sample and reverse transcribed into complementary DNA (cDNA). The cDNA is then amplified using specific primers and fluorescent probes that target specific regions of the SARS-CoV-2 genome, majorly the structural proteins. The results of the qPCR for SARS-CoV-2 detection are reported as cycle threshold (Ct) values, which indicate the presence or absence of the target genes [51]. The lower the Ct value, the higher the viral load in the clinical sample; a Ct value of less than 35 is considered positive for SARS-CoV-2, indicating the presence of viral RNA in the sample, while a low Ct value of less than 20 indicates a high viral load, while a high Ct value more than 30 indicates a low viral load.

This study explored the virucidal effect of biosynthesized silver nanoparticles against the SARS-CoV-2 virus at non-toxic concentrations using qPCR. To assess the dose-dependent effect of the synthesized AgNPmo on SARS-CoV-2 in a viral transport medium (VTM), the VTM was incubated with the biosynthesized AgNPmo over a period of 2 days at 24-hour intervals. The sample controls (SC) were included to confirm whether the increase in the Ct values is due to the activity of AgNPmo on the virus or environmental conditions, such as temperature, change in incubation time, or error in the viral RNA extraction or qPCR setup.

Figs 7 and 8 represent the cycle threshold (Ct) values of the target SARS-CoV-2 genes (ORF-Lab1ab and N-genes), respectively, against the AgNPmo dilutions. Starting with the IC50 concentration from the cytotoxicity assay, the AgNPmo had a similar progressive inhibitory effect at all the concentrations for both target genes. The inhibitory effect of the AgNPmo was also time-dependent, while the Ct values increased with an increase in the hours of incubation. The Ct values for the sample controls (SC) remained steady over the incubation period. A slight upward trend was

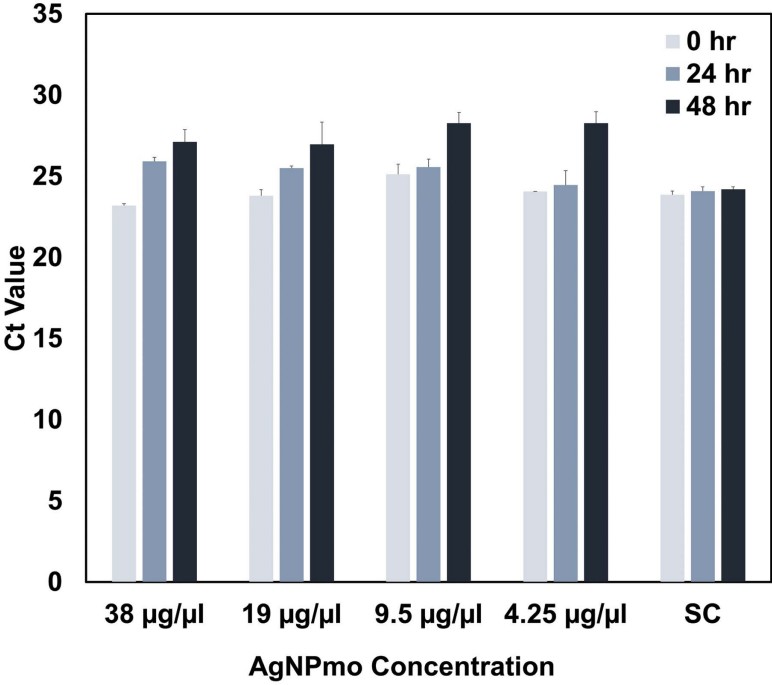

**Fig 7. Inhibitory effect of the biosynthesized nanoparticles against SARS-CoV-2 ORF-Lab 1 gene.** Sample control (SC) is the positive control. Bars represent mean±SEM of duplicate experiments. *p<0.05 compared to SC.

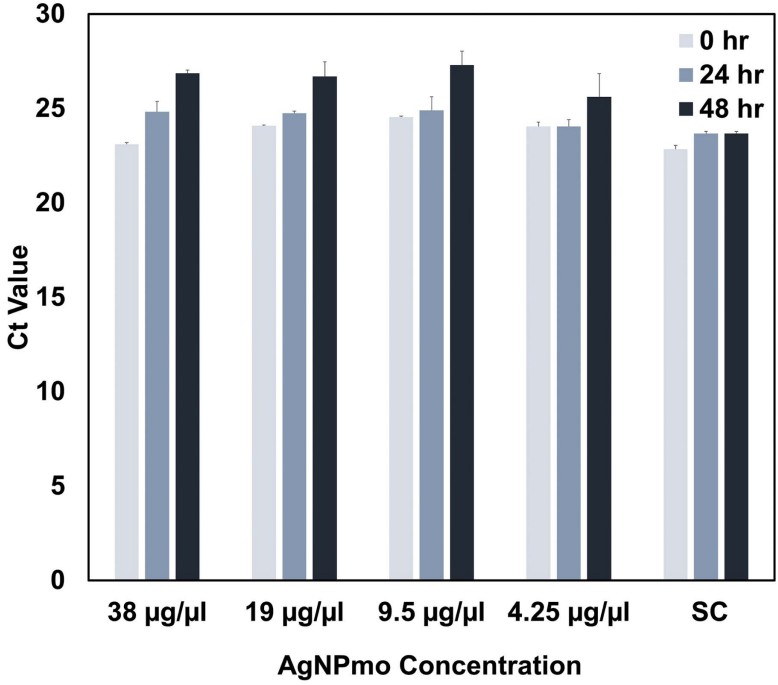

**Fig 8. Inhibitory effect of the biosynthesized nanoparticles against SARS-CoV-2 N-gene.** Sample control (SC) is the positive control. Bars represent mean ± SEM of duplicate experiments. *p < 0.05 compared to SC.

observed in the inhibitory effect of the AgNPmo with a decrease in the nanoparticle concentration for both genes (Fig 7 and 8). However, at 4.25 μg/mL concentration, there was no significant increase in the Ct values at 0 and 24 hours of incubation. The rapid increase in the inhibitory effect of AgNPmo on SARS-CoV-2 at 48 hours may be due to the long-time of incubation, which could have allowed the AgNPmo to interact well with the viral particles. The antiviral effect was evidenced by a progressive increase in Ct values with increasing incubation time. At 48 h, the Ct values rose significantly compared to controls, confirming inhibition of viral replication. This time- and dose-dependent effect provides direct evidence for the antiviral activity of AgNPmo.The most compelling result of our study is the increase in Ct values after 48 hours of exposure to AgNPmo, suggesting a strong inhibitory effect on SARS-CoV-2 replication. This observation supports the hypothesis proposed by Ratan et al. (2021) [32] and recently reviewed by Luceri et al. (2023) [20], who noted that AgNPs interfere with the integrity of viral envelopes, inhibit fusion with host membranes, and disrupt genome replication.

Therefore, the increase in Ct-values observed in this study may result from the AgNPmo interacting with viral replication, thereby inhibiting the viral RNA synthesis or assembly of viral particles. The AgNPmo might also have interfered with the structural proteins and inhibited their ability to bind with cell receptors or the genetic material of viruses, thereby inhibiting viral replication at the optimal concentration [52].

Baghban et al. (2024) [33] demonstrated that metallic nanoparticles can be particularly effective when allowed to undergo extended incubation with viral particles, supporting our time-dependent virucidal findings. The observation that lower concentrations (9.5 μg/mL) were still effective also suggests potential for safe application at low doses.

The beneficial effects of AgNPmo may be attributed to phytochemicals in *Moringa oleifera* leaves, particularly flavonoids and phenolic compounds, which serve as natural reducing and capping agents. These compounds enhance nanoparticle stability and may synergize with AgNPs by contributing to antioxidant and antimicrobial activities.

Mechanically, AgNPs are known to disrupt viral envelopes, interfere with receptor binding, and induce oxidative stress in microbial cells. Together, these effects explain the observed antimicrobial and antiviral outcomes.

Our findings are relevant in light of emerging evidence linking SARS-CoV-2 transmission to water systems. Studies have emphasized the need to identify affordable and scalable technologies for viral decontamination in wastewater. Our data demonstrate that AgNPmo is a viable candidate: it is low-cost, environmentally friendly, and exhibits potent virucidal effects without harming eukaryotic cells at effective concentrations.

In summary, this study showed that the biogenic silver nanoparticles were effective in inhibiting SARS-CoV-2. The virus inside the VTM showed an increase in Ct values, indicating a reduction in the virus's effectiveness. The results suggest that the nanoparticles are also relatively safe at controlled concentrations. The findings from this study are valuable and can be repurposed as a cost-effective and less toxic method of degrading SARS-CoV-2 particles in environmental water samples. Our approach, which utilizes inexpensive Moringa leaves and rapid microwave synthesis, aligns with reports that green AgNP production can cost approximately £8 per gram, compared to £18–28 per gram for commercial AgNPs [53]. This supports the potential cost-effectiveness of AgNPmo, though a direct economic evaluation in our system remains necessary.

While this study provides promising insights into the antiviral properties of *Moringa*-synthesized silver nanoparticles (AgNPmo), it has some limitations: 1) all experiments were conducted under *in vitro* conditions using Vero cells and viral transport medium, which do not fully replicate the complexity of real environmental or physiological systems. Consequently, the efficacy and safety of AgNPmo in actual water treatment settings or in vivo models have not been tested. 2) Although the increase in qPCR cycle threshold (Ct) values suggests viral inhibition, the exact molecular mechanism by which AgNPmo exerts its antiviral effect against SARS-CoV-2 was not elucidated. 3) This study was limited to a single viral strain, without evaluating other SARS-CoV-2 variants or unrelated enveloped viruses; this restricts the generalizability of the findings. 4) While cytotoxicity was assessed in mammalian cells, no environmental toxicity studies were performed to evaluate the potential risks of AgNPmo release into ecosystems. 5) The cytotoxicity data in this study were obtained using the CCK-8 colorimetric assay, which measures cellular metabolic activity rather than direct cell counts. Metallic nanoparticles can interfere with colorimetric reagents by absorbing/scattering light or chemically interacting with the dye; to mitigate this, we included NP-only control wells and subtracted their absorbance from treated wells. Negative corrected viability values observed at very high nanoparticle concentrations likely reflect a combination of severe cytotoxicity and assay interference; future work should confirm these observations with orthogonal assays (LDH release, ATP assays) and microscopy. 6) Finally, although UV-Vis data indicated some long-term absorbance stability, a comprehensive analysis of shelf-life and storage conditions for practical deployment was not conducted. These limitations highlight the need for further in vivo, environmental, and mechanistic studies prior to field application.

## Materials and methods

### Collection of Plant material and preparation of extract

Fresh *Moringa oleifera* leaves were obtained from the gardens of the Redeemer's University, Ede, Osun State, Nigeria. Mr. Olusegun Adedeji of the Biological Sciences Department at the Redeemer's University identified and authenticated the plant leaf material, and a voucher specimen (MO-01/2023) was put in the herbarium for referencing. The leaves were rinsed thoroughly with distilled and deionized water to remove dust particles. 20g of fresh leaf samples were submerged in 100 mL of deionized water. The mixture was heated using a microwave for 1 minute until a yellow-green coloration was observed. The crude extract was allowed to cool for 10 minutes, then filtered using Whatman® Number 1 filter paper. The filtered leaf extract was then stored in a refrigerator at 4°C for further use.

### Green synthesis of silver nanoparticles

The silver nanoparticles were synthesized using a mixture of the aqueous leaf extract and 10 mM silver nitrate (AgNO$_3$) solution (BDH Chemicals, UK) prepared in deionized water. The constant microwave irradiation method was used for the

nanoparticle synthesis [54]. Briefly, AgNO$_3$ solution and leaf extract were mixed at a ratio of 9:1, respectively. The solution was covered using a microwave-safe cling wrap and irradiated in a microwave oven operating at 400 watts for 30 seconds. The solution was allowed to stand for 5 minutes until reddish-brown coloration was observed. The overall silver nanoparticle synthesis (AgNPmo) progressed from a light yellow to a dark reddish-brown coloration.

The AgNPmo were purified by centrifugation at 1,500 rpm for 15 minutes at 4°C to avoid damaging the nanoparticles or altering their physicochemical properties. The supernatant was then discarded; the pellet was washed with deionized water, and the process was repeated. The resultant paste was dried overnight in a hot air oven at 25°C.

## Physicochemical characterization of silver nanoparticles

After visible coloration, the formation of the nanoparticle was first analyzed using UV-Vis spectrophotometry. The absorption spectra of the nanoparticles were analyzed within 350–500 nm wavelength using JENWAY 7305a spectrophotometer (Bibby Scientific, UK). The dried nanoparticle was diluted with distilled water, and UV-Vis spectrophotometry was repeated using nanodrop 2000 (Thermofisher, UK) after one year of synthesis. A Fourier Transform Infrared (FT-IR) spectrophotometer (Bruker, UK) and a KBr pellet method were used to determine biomolecules or functional groups responsible for the reduction and capping of the prepared AgNPmo nanoparticles within the spectral range of 3500–500 cm$^{-1}$.

X-ray Diffraction (XRD) analysis was performed using a D2 phaser x-ray diffractometer (Bruker, UK) with a Cu Kα radiation source in the 10°-60° 2θ range, 0.02°/s scanning step size. A smooth powder surface was placed in the sample holder with the primary divergent slit confirmed as 1 mm before closing the machine. The XRD machine was closed to avoid contact with the emitted radiation. The data was then transferred to the ICDD PDF-4 + software for phase analysis. Using Scanning Electron Microscopy (SEM) to estimate the dimensions of AgNPmo, the grain morphology was collected using an FEI Inspect F Scanning Electron Microscope (TSS Microscopy, USA). Before SEM analyses, the sample was coated with carbon to make the surface conductive to enable a better signal and good image under the microscope. The SEM was performed at an accelerating voltage of 10–15.0 kV, a spot size of 3.5–4.0, and a variable working distance.

## Antibacterial activity

Antibacterial activities of the AgNPs extract of *Moringa oleifera* leaves were carried out using agar well diffusion and the Minimum Inhibitory Concentration (MIC) method on *Pseudomonas aeruginosa* (ATCC 154423) and *Staphylococcus aureus* (ATCC 209233) clinical isolates. The isolates were obtained from the stock in the microbiology department at Redeemer's University. Bacteria suspension of each isolate (1.5 x 10$^8$ cells/mL) was prepared and compared with 0.5 McFarland standard, which was aseptically swabbed on a Mueller Hinton Agar plate with a sterile swab stick. Diffusion wells were made using a sterile cork borer (6 mm in diameter) and placed into the agar plates. Different concentrations of the AgNPmo, 100% (1.68 mg/mL), 50% (0.84 mg/mL), 25% (0.42 mg/mL), and 12.5% (0.21 mg/mL) prepared with 1% DMSO in two-fold serial dilution. 100 μl of each concentration was dispensed into each well labeled accordingly. The experiment was carried out in duplicate, and the plates were placed in the refrigerator for 30 minutes to allow the extract to diffuse into the agar. The plates were then incubated for 24h at 37°C.

## Cytotoxicity assay

The Cell viability test of the synthesized AgNPmo was analyzed through a Cell Counting Kit-8 (CCK-8) on Vero cells. The Vero cells (2 × 105) were seeded into each well of a 96-well plate and cultured for 24 hours before any treatment. The AgNPmo treatment groups were administered serial dilutions of the stock solution from 5 mg/mL to 0 mg/mL.10 μl CCK-8 reagent was added, and after 3 hours of incubation under humidified conditions at 37°C, the color development of the samples was measured at a test wavelength of 450 nm and a reference wavelength of 630 nm using an 800 TS microplate reader (Biotek Instruments, USA). The experiment was done in duplicate.

The relative cell viability was calculated as follows:

$$\text{Viability \%} = (\text{OD sample}/\text{OD control}) \times 100$$

The half maximal inhibitory concentration (IC50) was calculated using the statistics software Graph Prism version 9.

### Quantitative RT-PCR assay for antiviral activity

The COVID-19-positive samples preserved inside the Viral Transport Medium (VTM) were used for the experiment. The samples used were obtained from the Nigeria Institute of Medical Research (NIMR), and the ethical approval (IRB/23/029) was obtained from the ethical committee of the institute's reviewers' board. Verbal consent was obtained from the patient whose SARS-CoV-2-positive sample was used for this study.

AgNPmo stock concentration was diluted serially with sterile VTM to make 38, 19, 9.5, and 4.75 µg/mL based on the $IC_{50}$ report from the cytotoxicity assay. 100 µl of each dilution was mixed with 100 µl SARS-CoV-2 contaminated VTM and incubated for two days (0, 24, and 48 hours) at 25°C with periodic shaking. Viral RNA was extracted using the Qiagen RNA nucleic acid extraction kit (Qiagen, Hilden, Germany) at 0-, 24-, and 48-hour intervals. 60µl of the viral RNA was eluted after dry spinning the column for 2 minutes.

The quantitative RT-PCR was performed using the SCODA SARS-CoV-2 Fast PCR assay protocol according to Shaibu et al. (2023) [55]. The final reaction mixture of 25µl contained 7µl of the SCODA reagent A, 13µl of the SCODA reagent B, and 5µl of each sample. PCR amplification was achieved using a QuantStudio™ 5 Real-Time PCR System (ThermoFisher Scientific, UK) at optimized PCR conditions of 52°C for 10 minutes, 95°C for 10 seconds, 95°C for 5 seconds (40 cycles) and 56°C for 30 seconds (40 cycles). The reaction was performed in duplicate. Three controls were set up: First, the sample control (SC) contained the reaction mixture without AgNPmo. The negative control (NC) contained the reaction mixture and RNAse-free water without the sample. The third control contained the qPCR kit positive control (PC), replacing the sample. Both the NC and PC are used to control the quality of the PCR process.

### Inclusivity in global research

Additional information regarding the ethical, cultural, and scientific considerations specific to inclusivity in global research is included in the Supporting Information (S8 Checklist).

### Statistical analysis

Arithmetic mean and standard error were calculated for the antimicrobial and antiviral experiments. Statistical analysis was performed using GraphPad Prism version 9 (GraphPad Software, USA). One-way ANOVA was applied to compare treated groups with controls, and p-values $< 0.05$ were considered statistically significant.

### Supporting information

**S1 Fig. Showing negative control (NC) containing DMSO without the AgNPmo.**
(PDF)

**S2 Fig. Visual observation of AgNPmo synthesis showing the progressive color change of the reaction mixture (a) 0 min – yellowish prior to nanoparticle formation, (b) 5 min – brown indicating onset of nanoparticle formation, and (c) 10 min – reddish brown brown corresponding to complete AgNP formation.**
(PDF)

**S1 Table. Spectrophotometry readings.**
(PDF)

**S2 Table. Spectrophotometry readings.**
(PDF)

**S3 Table. Values for the cytotoxicity assay.**
(PDF)

**S4 Table. Zone of Inhibition (ZOI) of *P. aeruginosa* at different concentrations of the AgNPmo.**
(PDF)

**S5 Table. Zone of Inhibition (ZOI) of *S. aureus*at different concentrations of the AgNPmo.**
(PDF)

**S6 Table. Inhibitory effect of the biosynthesized nanoparticles against SARS-CoV-2 N-gene.**
(PDF)

**S7 Table. Inhibitory effect of the biosynthesized nanoparticles against SARS-CoV-2 ORF-Lab 1 gene.**
(PDF)

**S8 Checklist. Inclusivity in global research.**
(PDF)

**S9 Table. Major absorption features of AgNPmo identified from UV–Vis spectrophotometry.**
(PDF)

## Author contributions

**Conceptualization:** Adebayo J. Bello.

**Data curation:** Adebayo J. Bello, Omorilewa B. Ebunoluwa, Rukayat O. Ayorinde, Nneka Onyejepu, Joseph O. Shaibu, Adeniyi R. Adewole, Abeebat O. Adewole, Olusegun A. Adedeji, Olajumoke B. Oladapo.

**Formal analysis:** Adebayo J. Bello, Ololade O. Akinnusi, Olajumoke B. Oladapo, Temitope S. Popoola.

**Investigation:** Adebayo J. Bello, Omorilewa B. Ebunoluwa, Rukayat O. Ayorinde, Nneka Onyejepu, Joseph O. Shaibu, Adeniyi R. Adewole, Abeebat O. Adewole, Olusegun A. Adedeji, Oluwamodupe M. Arotiba.

**Methodology:** Adebayo J. Bello, Omorilewa B. Ebunoluwa, Rukayat O. Ayorinde, Nneka Onyejepu, Joseph O. Shaibu, Adeniyi R. Adewole, Abeebat O. Adewole, Olusegun A. Adedeji, Oluwamodupe M. Arotiba.

**Project administration:** Mujeeb O. Shittu, Adebayo J. Bello, Joseph B. Minari, Luqman A. Adams, Joy Okpuzor.

**Resources:** Adebayo J. Bello, Omorilewa B. Ebunoluwa, Joseph O. Shaibu, Abeebat O. Adewole, Temitope S. Popoola, Joseph B. Minari, Joy Okpuzor.

**Supervision:** Mujeeb O. Shittu, Adebayo J. Bello, Nneka Onyejepu, Joseph B. Minari, Luqman A. Adams, Joy Okpuzor.

**Validation:** Mujeeb O. Shittu, Adebayo J. Bello, Nneka Onyejepu, Joseph O. Shaibu, Luqman A. Adams, Joy Okpuzor.

**Visualization:** Mujeeb O. Shittu, Adebayo J. Bello, Joy Okpuzor.

**Writing – original draft:** Mujeeb O. Shittu, Adebayo J. Bello, Omorilewa B. Ebunoluwa, Rukayat O. Ayorinde.

**Writing – review & editing:** Mujeeb O. Shittu, Adebayo J. Bello, Omorilewa B. Ebunoluwa, Rukayat O. Ayorinde, Nneka Onyejepu, Joseph O. Shaibu, Adeniyi R. Adewole, Abeebat O. Adewole, Olusegun A. Adedeji, Ololade O. Akinnusi, Olajumoke B. Oladapo, Temitope S. Popoola, Oluwamodupe M. Arotiba, Joseph B. Minari, Luqman A. Adams, Joy Okpuzor.

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
