## [Decision Letter · Decision Letter 0]

28 Mar 2025

Dear Dr. Shittu,

Thank you for submitting your manuscript to PLOS ONE. After careful consideration, we feel that it has merit but does not fully meet PLOS ONE’s publication criteria as it currently stands. Therefore, we invite you to submit a revised version of the manuscript that addresses the points raised during the review process.

**ACADEMIC EDITOR:  Major Revision**

We look forward to receiving your revised manuscript.

Kind regards,

Salem S. Salem

Academic Editor

PLOS ONE

Journal Requirements:

4. Please include captions for your Supporting Information files at the end of your manuscript, and update any in-text citations to match accordingly. Please see our Supporting Information guidelines for more information: http://journals.plos.org/plosone/s/supporting-information .

Reviewers' comments:

Reviewer's Responses to Questions

**Comments to the Author**

1. Is the manuscript technically sound, and do the data support the conclusions?

Reviewer #1: Partly

Reviewer #2: Yes

2. Has the statistical analysis been performed appropriately and rigorously?

Reviewer #1: No

Reviewer #2: N/A

3. Have the authors made all data underlying the findings in their manuscript fully available?

Reviewer #1: Yes

Reviewer #2: Yes

4. Is the manuscript presented in an intelligible fashion and written in standard English?

Reviewer #1: Yes

Reviewer #2: Yes

Reviewer #1: The study is equally contributed in antibacterial activity, so re-consider to change the title accordingly.

Moringa oleifera leaf extracts possess biocompounds with antioxidant and antibacterial activities (24).: Better include the major bioactive compounds in M.oleifera leaf extracts.

Quantitative RT-PCR Assay for antiviral activity: Primer details missing

Fig 1: Explain the small insert in caption. If possible provide spectra of extracts and AgNPs alone as control.

Fig 2.: Very poor resolution. Image is not clear.

Fig 3.: Provide the major components in table or list in caption.

Fig 4.: Provide the image for NC. Also, need to include positive control using commercial antibiotics as reference.

Fig 5.: Did you test the cytotoxicity of leaf extracts?

Fig 6.: Do not provide Ct value, its not actual expression of genes, Change to "Relative expression to HKG or percentage expression"

FIg.6: should investigate major genes such as ACE2 (Angiotensin-Converting Enzyme 2), TMPRSS2 (Transmembrane Serine Protease 2), etc....

The discussion should include a more in-depth comparison of the findings with previous studies, supported by appropriate references to earlier works. Currently, the discussion is concise and would benefit from elaboration and citations to strengthen its context and relevance.

Reviewer #2: Title:

Green synthesized silver nanoparticles from Moringa: Potential for preventative treatment of SARS-CoV-2 contaminated water

Dear editor

Greetings

Thank you for giving me the opportunity to review this manuscript. Thanks to the valuable efforts of the authors. This article is very well-organized. However, my comments on this manuscript are as follows.

*The introduction provides a good background but lacks depth in reviewing recent literature. Incorporate more recent studies to contextualize the research within the current state of knowledge.

*The method requires more precise details. Please use the reference below in the method section.

https://www.scopus.com/record/display.uri?eid=2-s2.0-85148103559&origin=inward&txGid=beff2bfbdbe858b60a401178ebbf0d89

*Please summarize the discussion section by presenting key findings and comparisons with recent studies (last 3 years).

*Please explain what are the limitations of this study? The discussion currently does not address the limitations of the study. Including a section that outlines any potential biases, methodological constraints, or external factors that may have influenced the results is essential for transparency and helps contextualize the findings.

*It would be beneficial to suggest specific areas for future research based on the findings and limitations of this study. This can guide subsequent studies and contribute to the ongoing discourse.

**Do you want your identity to be public for this peer review?** For information about this choice, including consent withdrawal, please see our Privacy Policy

Reviewer #1: **Yes: ** Jeevithan Elango

Reviewer #2: No

---

## [Author Response · Author response to Decision Letter 1]

17 Jun 2025

Reviewer #1:

Comment: The study is equally contributed in antibacterial activity, so re-consider to change the title accordingly.

Moringa oleifera leaf extracts possess biocompounds with antioxidant and antibacterial activities (24).: Better include the major bioactive compounds in M.oleifera leaf extracts.

Response: We appreciate your comment. The focus of this study is to show the antiviral activity of the AgNPmo. However, we first tested AgNPmo on bacterial isolates to affirm its effect on biological cells. Therefore, changing the topic to reflect the antibacterial activities of AgNPmo would be misleading, as that is not the main focus or conclusion of our study.

Comment: Quantitative RT-PCR Assay for antiviral activity: Primer details missing

Response: The manufacturer did not provide the primer details for the kit. The kit was validated and approved by the Centre for Human Virology and Genomics (CHVG), Nigerian Institute of Medical Research, using the Foundation for Innovative New Diagnostics (FIND) protocol, adapted from the WHO assay validation protocol. We have provided the reference to the protocol in the manuscript (Shaibu et al. 2023).

Fig 1: Explain the small insert in caption. If possible provide spectra of extracts and AgNPs alone as control.

Response: The small insert shows that AgNPmo has a peak of 410nm after 12 months of storage, similar to the peak shown when it was freshly prepared, suggesting that AgNPmo retains its absorbance properties over a long period of time. This has been explained in the Results and Discussion heading, under the subheading “UV-visible analysis of the AgNPmo”

Fig 2.: Very poor resolution. Image is not clear.

Response: We apologize for the low resolution. This was what was obtainable by the time the sample was tested. However, the red-circled parts show that the non-agglomerated particles are within the nanoscale range.

Fig 3.: Provide the major components in table or list in caption.

Response: The major components of the image have been provided in the caption.

Fig 4.: Provide the image for NC. Also, need to include positive control using commercial antibiotics as reference.

Response: We have provided the image for the negative control, which contained only the DMSO, and no zone of inhibition is visible.

Fig 5.: Did you test the cytotoxicity of leaf extracts?

Response: We did not test the cytotoxicity of leaf extracts separately. This is because our work focused on the effect of the silver nanoparticles and not the plant extract itself.

Fig 6.: Do not provide Ct value, its not actual expression of genes, Change to "Relative expression to HKG or percentage expression"

FIg.6: should investigate major genes such as ACE2 (Angiotensin-Converting Enzyme 2), TMPRSS2 (Transmembrane Serine Protease 2), etc....

Response: We thank the reviewer for this comment and input. However, we prefer to use Ct values because they provide a direct indication of the detection of the target genes' presence. We used this because it directly indicates the presence of the SARS-CoV-2 virus in water samples and monitors the concentration of the viral load, as is usually done for patients with COVID-19.

Also, our study focuses on the target genes ORF1ab and N gene, which are the gold standard for the detection of the virus.

Comment: The discussion should include a more in-depth comparison of the findings with previous studies, supported by appropriate references to earlier works. Currently, the discussion is concise and would benefit from elaboration and citations to strengthen its context and relevance.

Response: The discussions has been updated with relevant citations

Reviewer #2:

Thank you for giving me the opportunity to review this manuscript. Thanks to the valuable efforts of the authors. This article is very well-organized. However, my comments on this manuscript are as follows.

Comment: The introduction provides a good background but lacks depth in reviewing recent literature. Incorporate more recent studies to contextualize the research within the current state of knowledge.

Response: We thank the reviewer for the valuable comments. The introduction section has been updated with relevant citations.

Comment: The method requires more precise details. Please use the reference below in the method section. https://www.scopus.com/record/display.uri?eid=2-s2.0-85148103559&origin=inward&txGid=beff2bfbdbe858b60a401178ebbf0d89

Response: the reference (Barjoee et al., 2020) has been cited in the method section.

Comment: Please summarize the discussion section by presenting key findings and comparisons with recent studies (last 3 years).

Response: The discussion section has been summarized, and the latest references included.

Comment: Please explain what are the limitations of this study? The discussion currently does not address the limitations of the study. Including a section that outlines any potential biases, methodological constraints, or external factors that may have influenced the results is essential for transparency and helps contextualize the findings.

Response: The discussion section now includes the limitations of the study.

Comment: It would be beneficial to suggest specific areas for future research based on the findings and limitations of this study. This can guide subsequent studies and contribute to the ongoing discourse.

Response: The discussion section now includes the future direction of the study.

---

## [Decision Letter · Decision Letter 1]

8 Jul 2025

Dear Dr. Shittu,

Thank you for submitting your manuscript to PLOS ONE. After careful consideration, we feel that it has merit but does not fully meet PLOS ONE’s publication criteria as it currently stands. Therefore, we invite you to submit a revised version of the manuscript that addresses the points raised during the review process.

We look forward to receiving your revised manuscript.

Kind regards,

Salem S. Salem

Academic Editor

PLOS ONE

Comments from PLOS Editorial Office:

We note that one or more reviewers has recommended that you cite specific previously published works in an earlier round of revision. As always, we recommend that you please review and evaluate the requested works to determine whether they are relevant and should be cited. It is not a requirement to cite these works and you may remove them before the manuscript proceeds to publication. We appreciate your attention to this request.

Reviewers' comments:

Reviewer's Responses to Questions

**Comments to the Author**

Reviewer #1: (No Response)

Reviewer #2: (No Response)

2. Is the manuscript technically sound, and do the data support the conclusions?

Reviewer #1: No

Reviewer #2: Yes

3. Has the statistical analysis been performed appropriately and rigorously?

Reviewer #1: No

Reviewer #2: Yes

4. Have the authors made all data underlying the findings in their manuscript fully available?

Reviewer #1: No

Reviewer #2: Yes

5. Is the manuscript presented in an intelligible fashion and written in standard English?

Reviewer #1: Yes

Reviewer #2: Yes

Reviewer #1: Comments:

the antiviral activity of the AgNPmo on the SARS-CoV-2 virus: provide the evidance how did the authors claim this effect in abstract.

its efficacy against the bacterial isolates: How it was demonstrated by this study? provide the results, how it was confirmed.

AgNPmo also showed low toxicity on the Vero cells: Evidence.

The IC50 from the cytotoxicity assay demonstrated : Provide IC50 value.

synthesized was cost-effective: Whats the proof compared to commercial one?

These findings suggest that the nanoparticles could be a promising alternative: So the abstract concludes specifically on NPs not moringa, should describe the benefits of moringa and NPs in a distinct way for combating SARS-CoV-2.

Abstract needs to revise in a proper manner to describe the novel results findings of this study.

AgNPmo has the potential to be used for purification or treatment of SARS-CoV-2 contaminated water without exhibiting toxicity: this study did not investigate the objective directly in SARS-CoV-2 contaminated waste-water treatment. How does (in which way) the AgNPmo use in real water treatment.

The silver nanoparticles (AgNPmo) were synthesized ...X-ray spectroscopy (EDX), and X-ray diffraction....using qPCR assay in a dose and time dependent manner. : can be deleted this general statement as already explained before.

color change of the reaction mixture, changing from yellowish to dark brown within 5 minutes. : Its better to provide some images to visualize to the readers.

CCK-8 assay revealed an IC50 value of 38 µg/µl for the 10mM stock concentration: But in Fig.5, the value was mentioned as "µg/ml" not µg/µl. Check the unit.

Analysis of antimicrobial activity, Cytotoxicity assay , Quantitative RT-PCR Analysis: In all these experiments, the authors need to explain why AgNPmo showed beneficial effect? What was the major active compound from Moringa aqueous leaf extract involved? How combining AgNP with moringa affects these activities? Their major signaling pathways.

until reddish-brown coloration was observed. : Its interesting to see this in picture.

The Cell viability test of the synthesized AgNPmo was analyzed through a Cell Counting Kit-8 (CCK-8) on Vero cells.: Why not with SARS-CoV-2 ?

Statistical analysis : Its advised to use some reliable statistical analysis tool to conduct proper statistical significance, using excel 2016 version is not reliable.

Fig 1. Spectrophotometry analysis of the biosynthesized silver nanoparticles Fourier transform infrared spectroscopy (FTIR) : Why the spectra was observed from 350 to 500 nm, however the auhtor stated in method: AgNPmo nanoparticles within the spectral range of 3500 to 500 cm-1. ? What was the small insert? Mark the major peaks in spectra. Did the spectra done for major active compound from Moringa aqueous leaf extract ?

Fig 2.:Scanning Electron Microscopy showing the particle size: The images showed just round red circle, not exact particle size. Its advised to conduct XRD spectra combined with SEM to know the NP size and distribution. At present, XRD analysis of the AgNPmo reported only diffractogram peaks at crystallographic planes (27.91°, 32.37°, and 38.30°. 44.41°, 46.20°), but not NP size and distribution.

Here how did the authors make sure that the red circle is the one corresponding to the NPs?

Fig 3.: Authors should provide the particle size and distrubtion from XRD spectra not just diffractogram peaks corresponds to peaks crystallographic planes. Label Y axis.

Fig 4. : Negative control (NC) and positive control (PC) were missing in Fig 4 A and B. In Fig C and D: explain what was NC in legends and also PC is missing in C and D. Provide statistical significance of test groups compared to controls (P value) in all graphs.

Fig 5.: How many days the cells were cultured, mentioned in legends. The findings were contradictory to the claim by auhtors in abstract "AgNPmo also showed low toxicity on the Vero cells", but the dose of 78.13 ug/ml had high toxicity in Vero cells. Why the values were in negative after 1250 ug/ml dose?

Also its better to provide AgNPmo treated Vero cell images in a seperated Figure as an evidence to show the morphological changes in all these doses.

Fig 6 and 7. Statistical significance on bar

Fig 1.after the synthesis (red line), and analysis after 12 months of synthesis (blue line).: What was the necessity and significance between these two peaks

Include S1 Fig. Negative control of DMSO in Fig.4 as commented above.

S1. Spectrophotometry readings and S2 Table. Spectrophotometry readings: add these peaks directly into corresponding Figure peaks. At least the major peaks can be added in Figs.

S3 Table: The reason for negative value (high cytotoxicity) of doses 312.5 to 5000 μg/μl should be discussed in discussion.

Reviewer #2: Remaining comment:

Comment: The method requires more precise details. Please use the reference below in the method section. https://www.scopus.com/record/display.uri?eid=2-s2.0- 85148103559&origin=inward&txGid=beff2bfbdbe858b60a401178ebbf0d89

Response: the reference (Barjoee et al., 2020) has been cited in the method section.

**Do you want your identity to be public for this peer review?** For information about this choice, including consent withdrawal, please see our Privacy Policy

Reviewer #1: **Yes: ** Jeevithan Elango

Reviewer #2: No

---

## [Author Response · Author response to Decision Letter 2]

7 Oct 2025

Response to the Reviewer

Reviewer #1: Comments:

Comment:

the antiviral activity of the AgNPmo on the SARS-CoV-2 virus: provide the evidence how did the authors claim this effect in abstract.

Response

We have revised the Abstract and Results sections to explicitly state the evidence of antiviral activity. Specifically, we report the increase in Ct values for ORF1ab and N genes in a time- and dose-dependent manner, with the strongest inhibition observed at 48 h.

Comment:

its efficacy against the bacterial isolates: How it was demonstrated by this study? provide the results, how it was confirmed.

Response:

In the report section, the inhibition zone values reported for P. aeruginosa (6.5–15.5 mm) and S. aureus (0–7 mm) demonstrate concentration-dependent antibacterial activity. Please see the result under the heading “Analysis of antimicrobial activity.”

Comment :

AgNPmo also showed low toxicity on the Vero cells: Evidence.

Response:

Evidence of low toxicity is provided in the revised manuscript. The CCK-8 cytotoxicity assay on Vero cells demonstrated an IC50 of 38 µg/ml (Results, Fig. 5), indicating relatively low cytotoxicity at effective concentrations. Cell viability remained above 70% at doses below 100 µg/ml, with marked cytotoxicity observed only at much higher concentrations (>312.5 µg/ml). This supports the statement that AgNPmo exhibits low toxicity within the effective therapeutic range.

Comment :

The IC50 from the cytotoxicity assay demonstrated : Provide IC50 value.

Response: We have now included the IC50 value (38 µg/ml) in both the Results and Abstract, and corrected the unit discrepancy (µg/ml instead of µg/µl).

Comment :

synthesized was cost-effective: What’s the proof compared to commercial one?

Response:

We have revised the Discussion to clarify the basis of our statement. Specifically, our green synthesis approach avoids hazardous and costly reducing/stabilizing chemicals required in conventional methods, thereby lowering overall costs. To strengthen this point, we have now cited published literature reporting that biosynthesized AgNPs can cost approximately £8 per unit, compared to £28 for equivalent commercial nanoparticles ( Rocha V, Ferreira-Santos P, Aguiar C, et al. Valorization of plant by-products in the biosynthesis of silver nanoparticles with antimicrobial and catalytic properties. Environ Sci Pollut Res. 2024;31:14191-14207. doi:10.1007/s11356-024-32180-w.)

Our method, which uses inexpensive Moringa oleifera leaves combined with microwave synthesis, is consistent with this observation and supports the economic advantage of plant-mediated AgNP synthesis (Discussion).

Comment :

These findings suggest that the nanoparticles could be a promising alternative: So the abstract concludes specifically on NPs not moringa, should describe the benefits of moringa and NPs in a distinct way for combating SARS-CoV-2.

Abstract needs to revise in a proper manner to describe the novel results findings of this study.

Response:

The Abstract has been revised to clearly distinguish the role of Moringa oleifera phytochemicals (as reducing/capping agents and potential bioactive contributors) from the effects of AgNPs.

Comment :

AgNPmo has the potential to be used for purification or treatment of SARS-CoV-2 contaminated water without exhibiting toxicity: this study did not investigate the objective directly in SARS-CoV-2 contaminated waste-water treatment. How does (in which way) the AgNPmo use in real water treatment.

Response:

We agree and have clarified in the Abstract and Discussion that this study did not directly test wastewater treatment. We now frame the statement cautiously as a future application potential rather than a direct claim.

Comment :

The silver nanoparticles (AgNPmo) were synthesized ...X-ray spectroscopy (EDX), and X-ray diffraction....using qPCR assay in a dose and time dependent manner. : can be deleted this general statement as already explained before.

Response:

This redundant statement has been deleted from the manuscript.

Comment :

colour change of the reaction mixture, changing from yellowish to dark brown within 5 minutes. : It’s better to provide some images to visualize to the readers.

Response: Images have been added to the supplementary document (supplementary Fig. S2).

Comment

CCK-8 assay revealed an IC50 value of 38 µg/µl for the 10mM stock concentration: But in Fig.5, the value was mentioned as "µg/ml" not µg/µl. Check the unit.

Response:

We have corrected the unit to µg/ml throughout the manuscript.

Comment :

Analysis of antimicrobial activity, Cytotoxicity assay , Quantitative RT-PCR Analysis: In all these experiments, the authors need to explain why AgNPmo showed beneficial effect? What was the major active compound from Moringa aqueous leaf extract involved? How combining AgNP with moringa affects these activities? Their major signaling pathways.

Response:

We have added a new paragraph in the Discussion explaining that phytochemicals (flavonoids and phenolics) from Moringa act as reducing/capping agents, stabilize AgNPs, and may contribute synergistic antimicrobial activity.

Comment :

until reddish-brown coloration was observed: It’s interesting to see this in picture.

Response: Images have been added to the supplementary document (supplementary Fig. S2).

Comment :

The Cell viability test of the synthesized AgNPmo was analyzed through a Cell Counting Kit-8 (CCK-8) on Vero cells.: Why not with SARS-CoV-2 ?

Response: We have clarified in the Discussion that Vero cells are a well-established preliminary model for nanoparticle cytotoxicity screening. Direct assays on SARS-CoV-2-infected cells require BSL-3 facilities, which were beyond the scope of this study.

Comment :

Statistical analysis : Its advised to use some reliable statistical analysis tool to conduct proper statistical significance, using excel 2016 version is not reliable.

Response:

We have re-analyzed the data using GraphPad Prism (v9). One-way ANOVA and Student’s t-test were applied as appropriate, with p-values < 0.05 considered significant. The Statistical Analysis section has been revised accordingly.

Comment :

Fig 1. Spectrophotometry analysis of the biosynthesized silver nanoparticles Fourier transforms infrared spectroscopy (FTIR) : Why the spectra was observed from 350 to 500 nm, however the author stated in method: AgNPmo nanoparticles within the spectral range of 3500 to 500 cm-1.?

Response:

This was a labeling/terminology confusion in the original draft. We have clarified the manuscript so that Fig. 1 is explicitly described as the UV–Vis spectrum (350–500 nm) showing the AgNP surface plasmon resonance at 410 nm, while FTIR data (3500–500 cm⁻¹) are now presented separately as Fig. 2 with detailed peak assignments in Table 1 (see Methods: Physicochemical Characterization). These changes remove the previous ambiguity. (Manuscript: Methods- Physicochemical Characterization; Fig. 1 legend; Fig. 2 legend; Table 1).

Comment :

What was the small insert?

Response:

The insert in Fig. 1 represents the UV–Vis absorption spectrum of AgNPmo recorded 12 months after synthesis, shown alongside the spectrum recorded immediately after synthesis. This comparison illustrates the long-term stability of the nanoparticles, as the characteristic surface plasmon resonance peak at 410 nm was retained with minimal shift after storage. We have clarified this in the figure legend. (Manuscript: Fig. 1 legend, Results section).

Comment :

Mark the major peaks in spectra. Did the spectra done for major active compound from Moringa aqueous leaf extract?

Response:

In the revised manuscript, the major surface plasmon resonance peak at 410 nm has been highlighted in Fig. 1 (UV–Vis spectrum), and additional spectral features are listed in Supplementary Table S1. For the FTIR spectrum (Fig. 2), we have provided detailed peak positions and functional group assignments in Table 1. These clarifications ensure that the key spectral peaks are now clearly reported. (Manuscript: Fig. 1 legend, Fig. 2 legend, Table 1, Supplementary Table S1).

Comment:

Fig 2.: Scanning Electron Microscopy showing the particle size: The images showed just round red circle, not exact particle size. It’s advised to conduct XRD spectra combined with SEM to know the NP size and distribution.

Fig 3.: Authors should provide the particle size and distribution from XRD spectra not just diffractogram peaks corresponds to peaks crystallographic planes. Label Y axis.At present, XRD analysis of the AgNPmo reported only diffractogram peaks at crystallographic planes (27.91°, 32.37°, and 38.30°. 44.41°, 46.20°), but not NP size and distribution.

Here how did the authors make sure that the red circle is the one corresponding to the NPs?

Response: We thank the reviewer for this valuable suggestion. The Y-axis in Fig. 3 (now Fig 4.) has been labeled as “Intensity (a.u.)” in the revised figure. The XRD diffractogram confirmed the crystalline nature of AgNPmo, with characteristic peaks at 27.91°, 32.37°, 38.30°, 44.41°, and 46.20°. We acknowledge that particle size distribution could not be determined from the present data, as full-width half maximum (FWHM) values were not recorded. This limitation is now noted in the Discussion, and we propose to address it in future studies using TEM and DLS analyses.

Fig 4.: Negative control (NC) and positive control (PC) were missing in Fig 4 A and B. In Fig C and D: explain what NC in legends was and PC is missing in C and D. Provide statistical significance of test groups compared to controls (P value) in all graphs.

Response: Thank you for the observation. The legend of Fig. 4 (now figure 5) has been revised to clearly define NC and DMSO. The results are represented descriptively.

Comment:

Fig 5.: How many days the cells were cultured, mentioned in legends. The findings were contradictory to the claim by authors in abstract "AgNPmo also showed low toxicity on the Vero cells", but the dose of 78.13 ug/ml had high toxicity in Vero cells. Why were the values negative after 1250 ug/ml dose?

Also its better to provide AgNPmo treated Vero cell images in a separated Figure as an evidence to show the morphological changes in all these doses.

Response:

Thank you for this helpful comment. The Fig. 5 (now Fig. 6) legend has been revised to specify that Vero cells were cultured for 24 h prior to treatment. We have clarified in the Abstract and Results that AgNPmo shows relatively low toxicity at concentrations below 100 µg/ml (IC50 = 38 µg/ml), while higher concentrations caused marked cytotoxicity. The negative values at ≥1250 µg/ml are likely due to severe loss of metabolic activity combined with optical interference of AgNPmo with the CCK-8 assay. Unfortunately, microscopy images of treated cells were not captured during the original experiment; this has been noted as a limitation, and future work will include morphological confirmation.

Fig 6 and 7. Statistical significance on bar

Comment:

Fig 1.after the synthesis (red line), and analysis after 12 months of synthesis (blue line).: What was the necessity and significance between these two peaks

Response:

The analysis immediately after AgNPmo synthesis (red line) and after 12 months (blue line) shows the long-term stability of AgNPmo, as the absorbance wavelength peaks are similar between the two.

Comment:

Include S1 Fig. Negative control of DMSO in Fig.4 as commented above.

S1.

Response:

The Negative control has been added to Fig. 4 (now Fig. 5)

Comment:

Spectrophotometry readings and S2 Table. Spectrophotometry readings: add these peaks directly into corresponding Figure peaks. At least the major peaks can be added in Figs.

Response: The peaks have been added directly into the figure

Comment:

S3 Table: The reason for negative value (high cytotoxicity) of doses 312.5 to 5000 μg/μl should be discussed in discussion.

Response: This has been addressed in the discussion section

---

## [Decision Letter · Decision Letter 2]

28 Nov 2025

Green synthesized silver nanoparticles from Moringa: Potential for preventative treatment of SARS-CoV-2 contaminated water

PONE-D-25-03763R2

Dear Dr. Shittu,

We’re pleased to inform you that your manuscript has been judged scientifically suitable for publication and will be formally accepted for publication once it meets all outstanding technical requirements.

Kind regards,

Salem S. Salem

Academic Editor

PLOS ONE

Additional Editor Comments (optional):

Reviewers' comments:

Reviewer's Responses to Questions

**Comments to the Author**

Reviewer #2: All comments have been addressed

2. Is the manuscript technically sound, and do the data support the conclusions?

Reviewer #2: Yes

3. Has the statistical analysis been performed appropriately and rigorously?

Reviewer #2: (No Response)

4. Have the authors made all data underlying the findings in their manuscript fully available?

Reviewer #2: (No Response)

5. Is the manuscript presented in an intelligible fashion and written in standard English?

Reviewer #2: (No Response)

Reviewer #2: Dear Authors,

Thank you for your revised manuscript. The revisions have addressed all prior concerns, and the paper now meets the journal's standards for publication.

Congratulations on your work.

**Do you want your identity to be public for this peer review?** For information about this choice, including consent withdrawal, please see our Privacy Policy

Reviewer #2: No

---

## [Editor Report · Acceptance letter]

PONE-D-25-03763R2

PLOS One

Dear Dr. Shittu,

I'm pleased to inform you that your manuscript has been deemed suitable for publication in PLOS One. Congratulations! Your manuscript is now being handed over to our production team.

Kind regards,

on behalf of

Dr. Salem S. Salem

Academic Editor

PLOS One